# Assessment of Genetic Diversity and Protein Content of Scandinavian Peas (*Pisum sativum*)

Louise Winther, Søren Kjærsgaard Rasmussen , Gert Poulsen and Conny Bruun Asmussen Lange *

Department of Plant and Environmental Sciences, University of Copenhagen, Thorvaldsensvej 40,
DK-1871 Copenhagen, Denmark
* Correspondence: con@plen.ku.dk

**Abstract:** We produced homogeneous lines of 227 pea accessions from the Nordic Genetic Resource Center via single seed descent. The genetic diversity among these, mostly Scandinavian accessions, was investigated using three microsatellite markers, A9, AC58 and AA5. The microsatellites were highly informative and separated 153 of 194 accessions on a Neighbor Joining topology. The high polymorphism information content (*PIC*) values between 0.87 and 0.91 indicated that the gene bank material contains a large number of pea accessions with different breeding histories. The peas were grown in the field for two years and seed protein content showed variation between 9.3% and 34.1% over the years and accessions, respectively. The mean thousand seed weight was 152.05 g. More than 10 accessions had a protein content above 28%, showing that the collection has potential as breeding nursery for high-protein pea cultivars.

**Keywords:** pea; *Pisum sativum* L.; SSR markers; *PIC*; microsatellites; genetic diversity; Neighbor Joining tree; protein content; thousand seed weight

## 1. Introduction

Peas (*Pisum sativum* L.) have been grown in Denmark since 500–1500 AD and sweet and soft pea types for human consumption since 1600 [1]. *Pisum sativum*, originating in the Mediterranean region, has been adapted to the temperate climate and long days of Denmark. The world production of peas is 10.5 mill hectares [2]. *Pisum sativum* belongs to the legume family, Fabaceae, and is a self-pollinated, diploid species with 2n = 14 and a very large genome of 3.92 Gb [3]. Peas were the model plant used by Gregor Mendel to discover general rules for genetic heredity [4]. Peas are currently an important crop for food, feed and sidechain products, with a great potential for future farming. The content of dietary proteins in different pea cultivars ranges from 24.3 to 32.6%. The starch content is between 33.4 and 47.5% and peas contain fibers as well as mineral nutrients [5,6]. Others found that the protein content of pea cultivars varies with environmental and agronomic factors [7–9] with a typical span in protein content from 14.5% to 28% [10,11]. The symbiosis with nitrogen-fixing soil bacteria favors the growing of peas in low-input farming systems because it reduces the need for nitrogen fertilizers, and when used in crop rotation, peas add nitrogen to the soil [3,12].

The genetic diversity in peas has been studied using microsatellites also named simple sequence repeats (SSR). A worldwide collection of 372 pea cultivars and landraces has been analyzed using 29 different microsatellite markers [13]; a collection of 175 pea cultivars from the Czech Republic has been analyzed using seven microsatellite markers [14], and 130 pea landraces from Turkey were analyzed using 14 markers [15]. The genetic diversity measured as polymorphism information content (*PIC*) ranges from 0.15 to 0.96 for different types and numbers of microsatellites [15]. The microsatellite technique is also efficient when analyzing smaller dedicated pea collections such as garden peas, forage peas, landraces and historical collections from various countries, e.g., Tunisia, Anatolia, Australia, India, and from gene banks [16–25]. These studies included between 8 and 34 microsatellites showing

*PIC* values ranging from 0.19 to 0.84. Specific microsatellite markers are developed for a collection of field pea cultivars from Ethiopia, which also identify private alleles [26]. A recent study of a forage pea collection of landraces from Anatolia successfully differentiated the accessions into subgroups of locally adapted genotypes [17]. With the SSR markers already available in the literature cited above, the initial cost of employing SSR markers to the present study is low.

The polymorphic information content (*PIC*) describes the polymorphism and thus the informational value of the microsatellite. Generally, *PIC* values above 0.5 are highly informative, between 0.25 and 0.5 reasonably informative, and below 0.25 slightly informative [27]. In a study based on the pea genome, 309 microsatellites were constructed and 235 were polymorphic in either the first or second set of tests [28]. The number of PCR fragments (alleles) per microsatellite is between two and seven. The *PIC* value of the microsatellites varies from 0.04 to 1 with a mean of 0.63, thus showing a wide range of information from slightly to highly informative [27,28]. A comparison of the genetic variation and morphological features of 65 pea cultivars and 21 wild pea accessions [29] found 51% polymorphic information content and the dendrogram shows that diversity among the varieties from Europe is narrower than in the rest of the accessions. Furthermore, the study supports the assumption that the topology of the dendrogram reflects the pattern of refinement of the cultivars [29]. In an analysis of 19 pea cultivars primarily from Australia and Russia, five of eight microsatellites successfully produced 34 PCR fragments in total with 3 to 13 fragments per microsatellite [30]. The *PIC* values ranged from 0.18 to 0.79 with a mean of 0.62 and classified the microsatellites as highly informative. The genetic diversity among 35 pea accession was assessed with 15 microsatellites, which produced 41 different fragments, and all genotypes were identified on a UPGMA tree [31]. The *PIC* values ranged from 0.05 to 0.66 with a mean of 0.46. Jain et al. [32] investigated the genetic diversity among 96 cultivars from all over the world using 31 microsatellites along with 42 expressed sequence tag markers and 11 modern markers. The microsatellites identified 83 PCR fragments with an average of 2–6 per microsatellite and the *PIC* values varied from 0.01 to 0.56 with a mean of 0.29. Nasiri et al. [33] investigated 77 pea accessions from 17 countries using 10 microsatellites. The results included 59 PCR fragments, which varied from 2 to 8 per microsatellite. The *PIC* values ranged between 0.56 and 0.84, with a mean of 0.72. Kumari et al. [34] investigated 28 pea genotypes with 32 microsatellites, which resulted in 44 fragments with 2 to 4 fragments per microsatellites. The *PIC* values were between 0.31 and 0.66 with a mean of 0.49.

The current study surveyed a subset of the approximately 2500 pea accessions of Peas at the Nordic Genetic Resource Center, NordGen, https://www.nordgen.org/ (accessed on 16 March 2023), many of which were from the Weibullsholm *Pisum* collection. The accessions were primarily chosen based on prior knowledge of potential high protein content. The investigations were a first step in a pre-breeding process towards selecting and developing pea cultivars with a high protein content to meet the demand for plant-based protein for human consumption and as an alternative source of protein for husbandry. We address the following three questions: (1) What is the genetic diversity among the accessions and is the diversity structured? (2) What is the polymorphic information content? (3) What is the level of protein content and the thousand seed weight of accessions?

## 2. Materials and Methods

### 2.1. Plant Material

The plant material included in this study originated from the Nordic Genetic Resource Center, https://www.nordgen.org/ (accessed on 16 March 2023; Table 1. Most accessions originated from the Nordic countries and information on the specific location of origin, pedigree relationships, and proper cultivar name is in general sparse but some information may be retrieved in the seed database at https://www.nordic-baltic-genebanks.org/gringlobal/search.aspx (accessed on 16 March 2023). A total of 227 pea accessions were

grown from a single seed in pots in the greenhouse. Plants were grown to maturity and the seeds were collected for further studies in the field.

**Table 1.** List of *Pisum sativum* accessions included in this study. SKR is the working number given to the accession in this study. Accession is the number from NordGen, www.NordGen.org (accessed on 16 March 2023). Name included cultivar name, if existing, WBH designate a pre-breeding line from Weibuhlholm. AA5, AC58 and AA9 were the microsatellites used and x indicate that the accession produced a result. TSW were the thousand seed weight and then seed protein content in % for 2017, 2018 and 2019.

| SKR. No. | Accession | Name | AA5 | AC58 | AA9 | TSW 2017 (g) | Protein 2017 (%) | Protein 2018 (%) | Protein 2019 (%) |
|----------|-----------|------|-----|------|-----|--------------|------------------|------------------|------------------|
| SKR1 | NGB103441 | WBH 3441 | x | x | x | 178.1 | 23.4 | 26.9 | |
| SKR2 | NGB101418 | WBH 1418 | x | x | x | 107.3 | 20.3 | 20.8 | |
| SKR3 | NGB103563 | WBH 3563 | x | x | x | | 16.0 | | |
| SKR4 | NGB101515 | WBH 1515 | x | x | x | | | | |
| SKR5 | NGB100756 | WBH 756 | x | x | x | 185.0 | 25.0 | 29.9 | 25.0 |
| SKR6 | NGB101687 | WBH 1687 | x | x | x | | 19.7 | | |
| SKR7 | NGB105795 | multimicrodentatus | x | x | x | | 15.6 | | |
| SKR8 | NGB103423 | Latah | | x | x | | 14.3 | | |
| SKR9 | NGB100463 | WBH 463 | x | x | x | | 16.0 | | |
| SKR10 | NGB103565 | WBH 3565 | x | x | x | | 12.0 | | |
| SKR11 | NGB101735 | Improved Harbinger | x | x | x | 244 | 21.0 | 25.6 | |
| SKR12 | NGB101736 | rouge C15 | x | x | x | | 17.7 | | |
| SKR13 | NGB101339 | Bonneville | x | x | x | 180.2 | 26.4 | 22.0 | |
| SKR14 | NGB101741 | New Season | x | x | x | | 19.5 | | |
| SKR15 | NGB101603 | WBH 1603 | x | x | x | | | | |
| SKR16 | NGB103546 | W.S.U.-28 | x | x | x | | 22.3 | 25.9 | |
| SKR17 | NGB101772 | Wellensiek's tester | x | x | x | | 18.6 | | |
| SKR18 | NGB101325 | WBH 1325 | x | x | x | | | | |
| SKR19 | NGB101330 | WBH 1330 | x | x | x | | 20.0 | | |
| SKR20 | NGB101165 | WBH 1165 | x | x | x | 142.2 | 23.0 | 24.1 | |
| SKR21 | NGB103436 | WBH 3436 | x | x | x | | 18.3 | | |
| SKR22 | NGB102963 | Wilt Resistant Thomas Laxton | | | | 172.8 | 24.7 | 25.8 | |
| SKR23 | NGB101889 | | x | x | x | | | | |
| SKR24 | NGB103439 | WBH 3439 | x | x | x | | 13.9 | | |
| SKR25 | NGB103429 | WBH 3429 | x | x | x | | 20.6 | 24.1 | |
| SKR26 | NGB103583 | WBH 3583 | x | x | x | | | | |
| SKR27 | NGB100592 | WBH 592 | | x | x | | | | |
| SKR28 | NGB103452 | WBH 3452 | x | x | x | | | | |
| SKR29 | NGB102158 | WBH 2158 | x | x | x | 135.5 | 18.6 | 21.5 | |
| SKR30 | NGB103422 | Alaska | x | x | x | | 18.2 | | |

**Table 1.** *Cont.*

| SKR. No. | Accession | Name | AA5 | AC58 | AA9 | TSW 2017 (g) | Protein 2017 (%) | Protein 2018 (%) | Protein 2019 (%) |
|----------|-----------|------|-----|------|-----|--------------|------------------|------------------|------------------|
| SKR31 | NGB103449 | Feltham First | x | x | x | 178.9 | 21.5 | 24.4 | |
| SKR32 | NGB103426 | Juneau | x | x | x | | 17.8 | | |
| SKR33 | NGB103442 | WBH 3442 | x | x | x | | 14.4 | | |
| SKR34 | NGB103576 | WBH 3576 | x | x | x | 187 | 31.7 | | |
| SKR35 | NGB103578 | WBH 3578 | x | x | x | 193 | 25.6 | 26.1 | 20.5 |
| SKR36 | NGB100851 | procumbens | x | x | x | 165 | 23.4 | 28.5 | |
| SKR37 | NGB101338 | Salzmunder Edelperle | x | x | x | | 17.1 | | |
| SKR38 | NGB103433 | WBH 3433 | x | x | x | | 16.6 | | |
| SKR39 | NGB102177 | WBH 2177 | x | x | x | 132 | 23.7 | 27.3 | |
| SKR40 | NGB103438 | WBH 3438 | x | x | x | | 20.1 | | |
| SKR41 | NGB101608 | WBH 1608 | x | x | x | | | | |
| SKR42 | NGB103435 | WBH 3435 | x | | x | | 12.1 | | |
| SKR43 | NGB102058 | WBH 2058 | x | x | x | | 20.0 | | |
| SKR44 | NGB101524 | patelliformis | x | x | x | | | | |
| SKR45 | NGB101570 | WBH 1570 | x | x | x | | 13.1 | | |
| SKR46 | NGB103568 | WBH 3568 | x | x | x | 64.4 | 21.7 | 22.2 | |
| SKR47 | NGB103581 | WBH 3581 | x | x | x | | | | |
| SKR48 | NGB105136 | chlorotica | x | x | x | 76.4 | 30.0 | 22.9 | 24.4 |
| SKR49 | NGB102184 | New Era | x | x | x | | 16.6 | | |
| SKR50 | NGB102160 | WBH 2160 | x | x | x | 144.4 | 22.4 | 28.5 | |
| SKR51 | NGB103580 | WBH 3580 | x | x | x | 156 | 23.7 | 21.3 | |
| SKR52 | NGB102022 | WBH 2022 | x | x | x | 104 | 30.2 | 23.8 | |
| SKR53 | NGB102663 | WBH 2663 | x | x | x | | | | |
| SKR54 | NGB102203 | WBH 2203 | x | x | x | | 15.6 | | |
| SKR55 | NGB102217 | chlorotica | x | x | x | | 19.9 | | |
| SKR56 | NGB105350 | /chlorina | | | | 157.2 | 27.3 | 24.8 | |
| SKR57 | NGB102431 | Laxtonian | | | | 193 | 23.8 | 24.8 | 24.1 |
| SKR58 | NGB105124 | ageotropum | x | x | x | | 17.0 | | |
| SKR59 | NGB102496 | | x | x | x | 151 | 32.1 | | |
| SKR60 | NGB102214 | chlorotica | x | x | x | 123 | 23.3 | 28.9 | |
| SKR61 | NGB102210 | chlorotica | x | x | x | 145 | 23.8 | 24.3 | |
| SKR62 | NGB102136 | WBH 2136 | x | x | x | 162 | 25.4 | 26.3 | |
| SKR63 | NGB102216 | chlorotica | x | x | x | 104.8 | 29.4 | 30.6 | 24.3 |
| SKR64 | NGB105862 | densinodosum | | | | 155 | 20.9 | 34.1 | |
| SKR65 | NGB105428 | chlorotica | x | x | x | | 15.6 | | |
| SKR66 | NGB102574 | Beta | x | x | x | 150.1 | 22.4 | 24.6 | |
| SKR67 | NGB102622 | | x | x | x | 164 | 22.4 | 24.9 | 21.1 |
| SKR68 | NGB102988 | WBH 2988 | x | x | x | | 20.2 | | |
| SKR69 | NGB106051 | reduced in wax | x | x | x | 166.6 | 21.4 | 29.0 | |

**Table 1.** *Cont.*

| SKR. No. | Accession | Name | AA5 | AC58 | AA9 | TSW 2017 (g) | Protein 2017 (%) | Protein 2018 (%) | Protein 2019 (%) |
|---|---|---|---|---|---|---|---|---|---|
| SKR70 | NGB102432 | Hundredfold | x | x | x | | 18.2 | | |
| SKR71 | NGB105789 | /compactum | x | x | x | | 19.0 | | |
| SKR72 | NGB106060 | supaeromaculata | | | | | 16.3 | | |
| SKR73 | NGB102239 | | x | x | x | | 20.3 | | |
| SKR74 | NGB102369 | WBH 2369 | x | x | x | 129.2 | 21.9 | 25.2 | |
| SKR75 | NGB105051 | /compactum | x | x | x | | | | |
| SKR76 | NGB105432 | reductus | x | x | x | 84 | 26.2 | 25.1 | |
| SKR77 | NGB106080 | WBH 6080 | x | x | x | | 17.4 | | |
| SKR78 | NGB102579 | | x | | x | 231 | 27.0 | 23.1 | |
| SKR79 | NGB102581 | | x | x | x | | 24.5 | | |
| SKR80 | NGB105765 | chlorotica | | | x | 132 | 20.9 | 26.1 | |
| SKR81 | NGB103431 | WBH 3431 | x | x | x | | 18.1 | | |
| SKR82 | NGB102823 | Austrian Winter | x | x | x | | 13.1 | | |
| SKR83 | NGB103571 | WBH 3571 | x | x | x | | 14.4 | | |
| SKR84 | NGB103572 | WBH 3572 | x | x | x | | | | |
| SKR85 | NGB103573 | WBH 3573 | x | x | x | | 17.3 | | |
| SKR86 | NGB103585 | WBH 3585 | x | x | x | | 13.5 | | |
| SKR87 | NGB101452 | WBH 1452 | x | x | x | 159 | 22.3 | 22.0 | |
| SKR88 | NGB101192 | WBH 1192 | | | | 152 | 28.3 | 24.6 | 24.8 |
| SKR89 | NGB101391 | WBH 1391 | x | x | x | | | | |
| SKR90 | NGB101689 | WBH 1689 | x | x | x | | 25.1 | | |
| SKR91 | NGB101017 | WBH 1017 | x | x | x | 151.4 | 24.9 | 28.9 | |
| SKR92 | NGB101351 | WBH 1351 | x | x | x | | | | |
| SKR93 | NGB105534 | supaeromaculata 68 | x | x | x | 113.1 | 28.9 | 28.7 | 23.4 |
| SKR94 | NGB102188 | WBH 2188 | x | x | x | 181.87 | 22.6 | 27.7 | |
| SKR95 | NGB103420 | WA 788 | x | x | x | | 14.6 | | |
| SKR96 | NGB103434 | WBH 3434 | | | | | 18.3 | | |
| SKR97 | NGB103421 | Lilaska | | x | x | | 18.9 | | |
| SKR98 | NGB103428 | WBH 3428 | | | | | 18.5 | | |
| SKR99 | NGB103432 | WBH 3432 | | x | x | | 19.6 | | |
| SKR100 | NGB102070 | Puke | | x | x | | 16.2 | | |
| SKR101 | NGB103577 | WBH 3577 | x | x | x | 174.8 | 21.9 | 26.0 | |
| SKR102 | NGB100909 | WBH 909 | | | | 160 | 22.0 | 24.0 | |
| SKR103 | NGB101500 | WBH 1500 | | x | x | 99.1 | 20.5 | 26.3 | |
| SKR104 | NGB103437 | WBH 3437 | | | | 67 | 23.7 | 24.6 | |
| SKR105 | NGB102069 | Patea | x | x | x | 212.4 | 22.0 | 19.4 | |
| SKR106 | NGB103561 | WBH 3561 | | x | x | | 21.2 | | |
| SKR107 | NGB103567 | WBH 3567 | x | x | x | | 17.0 | | |
| SKR108 | NGB105310 | cerosa | x | x | x | 108.8 | 26.4 | 27.0 | 22.5 |

Table 1. *Cont.*

| SKR. No. | Accession | Name | AA5 | AC58 | AA9 | TSW 2017 (g) | Protein 2017 (%) | Protein 2018 (%) | Protein 2019 (%) |
|---|---|---|---|---|---|---|---|---|---|
| SKR109 | NGB103574 | WBH 3574 | x | x | x | | | | |
| SKR110 | NGB103575 | WBH 3575 | x | x | | | | | |
| SKR111 | NGB103579 | WBH 3579 | x | x | x | | 17.0 | | |
| SKR112 | NGB103584 | WBH 3584 | x | x | x | | | | |
| SKR113 | NGB103602 | WBH 3602 | x | x | x | | 11.2 | | |
| SKR114 | NGB103603 | WBH 3603 | x | x | x | | 18.7 | | |
| SKR115 | NGB102901 | | x | x | x | | 12.8 | | |
| SKR116 | NGB103430 | WBH 3430 | x | x | x | | 18.3 | | |
| SKR117 | NGB103427 | WBH 3427 | x | x | | | 17.1 | | |
| SKR118 | NGB102159 | WBH 2159 | x | x | | | 18.5 | | |
| SKR119 | NGB102844 | | x | | x | | 19.2 | | |
| SKR120 | NGB102185 | New Wales | x | x | x | 209 | 24.6 | 25.4 | 23.0 |
| SKR121 | NGB101132 | WBH 1132 | x | x | x | | | | |
| SKR122 | NGB103570 | WBH 3570 | x | x | x | | 19.5 | | |
| SKR123 | NGB103582 | WBH 3582 | | x | x | | 18.7 | | |
| SKR124 | NGB103604 | WBH 3604 | x | x | x | | 12.4 | | |
| SKR125 | NGB103605 | WBH 3605 | x | x | | | 13.3 | | |
| SKR126 | NGB101784 | Mrkos horizontale | x | x | x | | | | |
| SKR127 | NGB102063 | WBH 2063 | x | x | x | 172 | 24.4 | 26.9 | 20.7 |
| SKR128 | NGB105765 | chlorotica | | | | 125.1 | 26.1 | 25.1 | |
| SKR129 | NGB102578 | | x | | x | 148 | 33.6 | 26.0 | 21.2 |
| SKR130 | NGB102537 | | | | | | 19.1 | | |
| SKR131 | NGB103458 | | x | | x | 118.4 | 20.6 | 26.7 | 22.2 |
| SKR132 | NGB105161 | costata | x | x | x | 92 | 25.8 | 23.3 | |
| SKR133 | NGB102999 | WBH 2999 | x | x | x | | 17.4 | | |
| SKR134 | NGB102687 | WBH 2687 | x | x | | 178.5 | 21.2 | 26.9 | 22.7 |
| SKR135 | NGB105267 | variomaculata | x | | x | 148 | 27.3 | 29.8 | 22.0 |
| SKR136 | NGB105814 | chlorotica | x | x | x | 118 | 21.9 | 26.8 | |
| SKR137 | NGB102037 | WBH 2037 | x | x | x | | 17.2 | | |
| SKR138 | NGB102190 | WBH 2190 | x | | x | 223 | 28.8 | 26.1 | 22.4 |
| SKR139 | NGB102763 | Wonder Van Amerika I2048 | x | x | x | 142 | 22.2 | 27.4 | |
| SKR140 | NGB106000 | vixcerata | | | | 232 | 29.4 | 28.6 | 23.0 |
| SKR141 | NGB102582 | | | | | | 19.5 | | |
| SKR142 | NGB105449 | variomaculata | x | x | x | 120.2 | 22.9 | 27.5 | |
| SKR144 | NGB102370 | WBH 2370 | x | x | x | | 22.2 | | |
| SKR145 | NGB102621 | | x | x | x | | 19.2 | | |
| SKR146 | NGB102927 | | x | x | x | | 17.5 | | |
| SKR147 | NGB103459 | | x | x | x | 150 | 25.5 | 25.7 | |

**Table 1.** *Cont.*

| SKR. No. | Accession | Name | AA5 | AC58 | AA9 | TSW 2017 (g) | Protein 2017 (%) | Protein 2018 (%) | Protein 2019 (%) |
|---|---|---|---|---|---|---|---|---|---|
| SKR148 | NGB105820 | /xantha | x | x | x | | | | |
| SKR149 | NGB102521 | | x | x | x | 238 | 20.5 | 27.3 | 22.4 |
| SKR150 | NGB102617 | | | | | | 16.8 | | |
| SKR151 | NGB103457 | | x | x | x | 144 | 22.8 | 21.7 | |
| SKR152 | NGB105410 | variomaculata | x | x | x | 130.6 | 26.2 | 26.9 | 23.3 |
| SKR153 | NGB105981 | precocious yellowing | x | x | x | 107 | 24.5 | 29.1 | 23.6 |
| SKR154 | NGB102480 | | x | x | x | 124.4 | 25.5 | 29.5 | 21.4 |
| SKR155 | NGB102588 | | x | x | x | | | | |
| SKR156 | NGB105961 | chlorotica | x | x | x | 156.6 | 23.2 | 29.0 | |
| SKR157 | NGB102205 | chlorotica | x | x | x | 134 | 22.2 | 25.8 | |
| SKR158 | NGB102497 | | | | | 193 | 31.6 | | |
| SKR159 | NGB102533 | | | | | | 19.7 | | |
| SKR160 | NGB102831 | | x | x | x | 149 | 24.2 | 25.2 | |
| SKR161 | NGB103052 | WBH 3052 | x | x | x | 123 | 28.7 | 27.5 | 24.0 |
| SKR162 | NGB103061 | WBH 3061 | x | x | x | | | | |
| SKR163 | NGB103116 | | x | x | x | | 19.4 | | |
| SKR164 | NGB105261 | variomaculata | x | x | x | 150 | 24.3 | 33.1 | 20.0 |
| SKR165 | NGB105806 | chlorotica | x | x | x | 131 | 23.5 | 25.7 | 22.5 |
| SKR166 | NGB105983 | precocious yellowing | | | | 90 | 25.6 | 26.9 | 23.2 |
| SKR167 | NGB102005 | Nischkes Riesengefärgs Wi. Erbse | x | x | x | | 17.2 | | |
| SKR168 | NGB100993 | WBH 993 | x | x | x | 173 | 27.9 | 32.0 | 19.9 |
| SKR169 | NGB103559 | WBH 3559 | x | x | x | | 12.7 | | |
| SKR170 | NGB102405 | Supergrade | x | x | x | 194 | 21.9 | 24.6 | 21.8 |
| SKR171 | NGB101742 | New Wales | x | x | x | 182 | 22.5 | 29.9 | |
| SKR172 | NGB101888 | | x | x | x | | | | |
| SKR173 | NGB102423 | Senator | x | x | x | | 18.4 | | |
| SKR174 | NGB103560 | WBH 3560 | | | | | 10.9 | | |
| SKR175 | NGB102429 | Gradus | | | | | 19.7 | | |
| SKR176 | NGB103569 | WBH 3569 | x | x | x | | 12.9 | | |
| SKR177 | NGB103545 | Puget | x | x | x | | | | |
| SKR178 | NGB103566 | WBH 3566 | x | x | x | | 16.2 | | |
| SKR179 | NGB103484 | Dhamar | x | x | x | | | | |
| SKR180 | NGB103425 | WBH 3425 | x | x | x | | 18.7 | | |
| SKR181 | NGB103547 | Grant | x | x | x | 145 | 22.4 | 25.6 | |
| SKR182 | NGB102130 | WBH 2130 | x | x | | | 9.3 | | |
| SKR183 | NGB103609 | WBH 3609 | | | | | 17.2 | | |
| SKR184 | NGB101979 | Ambrosia | x | x | x | 144 | 24.0 | 27.4 | 21.8 |

| SKR. No. | Accession | Name | AA5 | AC58 | AA9 | TSW 2017 (g) | Protein 2017 (%) | Protein 2018 (%) | Protein 2019 (%) |
|---|---|---|---|---|---|---|---|---|---|
| SKR185 | NGB100800 | Primus | x | x | x | 149 | 22.5 | 26.8 | |
| SKR186 | NGB101721 | Midfreezer | x | x | x | | 19.1 | | |
| SKR187 | NGB101395 | WBH 1395 | x | x | x | 148 | 24.4 | 27.1 | 23.1 |
| SKR188 | NGB101677 | Mexique 4 | x | x | x | | 16.7 | | |
| SKR189 | NGB103440 | WBH 3440 | x | x | x | 143 | 22.7 | 26.8 | |
| SKR190 | NGB101462 | WBH 1462 | x | x | x | | 25.0 | 24.2 | |
| SKR191 | NGB101362 | clavicula | x | x | x | 156 | 17.0 | | |
| SKR192 | NGB100640 | WBH 640 | x | x | x | | 18.4 | | |
| SKR193 | NGB103450 | Meteor | x | x | x | | 18.6 | | |
| SKR195 | NGB103562 | WBH 3562 | x | x | x | 164 | 20.9 | 26.5 | |
| SKR196 | NGB101463 | Sigyn | x | x | x | 132 | 23.2 | 24.9 | |
| SKR198 | NGB101341 | Klema Vereduna | x | x | x | | 20.1 | | |
| SKR199 | NGB103424 | WBH 3424 | x | x | x | | 16.2 | | |
| SKR200 | NGB103544 | Ranger | x | x | | 214 | 21.7 | 25.2 | |
| SKR201 | NGB103564 | WBH 3564 | x | x | x | | 16.9 | | |
| SKR202 | NGB103610 | WBH 3610 | x | | x | | 15.7 | | |
| SKR203 | NGB105454 | viridis | x | x | x | 161 | 22.8 | 25.3 | |
| SKR204 | NGB105995 | vixcerata | x | x | x | 134 | 29.0 | 28.8 | 26.4 |
| SKR205 | NGB102212 | chlorotica | x | x | x | | 18.7 | | |
| SKR206 | NGB102688 | WBH 2688 | x | x | x | | | | |
| SKR207 | NGB105271 | costata | x | x | x | | 17.2 | | |
| SKR208 | NGB105565 | chlorotica | x | x | x | | 23.7 | | |
| SKR209 | NGB105848 | fasciata | x | x | x | | 26.0 | | |
| SKR210 | NGB106116 | narrow leaflet base | x | x | x | | 21.1 | | |
| SKR211 | NGB102178 | Thomas Laxton | x | x | x | 225 | 22.0 | 33.0 | |
| SKR212 | NGB102737 | WW 709 | x | x | x | | 17.0 | | |
| SKR213 | NGB102071 | Piri | x | x | x | | 18.6 | | |
| SKR214 | NGB102183 | Darkskin Perfection | x | x | x | 154 | 22.2 | 25.2 | |
| SKR215 | NGB103451 | Lilaska | x | x | x | | 16.7 | | |
| SKR216 | NGB102072 | Pania | x | x | | 175 | 20.8 | 27.4 | |
| SKR217 | NGB103606 | WBH 3606 | x | x | x | | | | |
| SKR218 | NGB103607 | WBH 3607 | | x | x | 254.2 | 23.9 | 26.3 | |
| SKR219 | NGB103628 | ramosus | x | x | x | 96 | 21.6 | 25.1 | |
| SKR220 | NGB101304 | WBH 1304-1 | | | | 159 | 22.4 | 27.9 | |
| SKR221 | NGB101304 | WBH 1304-2 | | | | | 16.1 | | |
| SKR222 | NGB101304 | WBH 1304-3 | | | | 150 | 22.4 | 27.7 | |
| SKR223 | NGB101304 | WBH 1304-4 | | | | | 18.3 | | |
| SKR224 | NGB101304 | WBH 1304-5 | | | | | 14.6 | | |

**Table 1.** *Cont.*

| SKR. No. | Accession | Name | AA5 | AC58 | AA9 | TSW 2017 (g) | Protein 2017 (%) | Protein 2018 (%) | Protein 2019 (%) |
|---|---|---|---|---|---|---|---|---|---|
| SKR225 | NGB101304 | WBH 1304-6 | | | | | 19.1 | | |
| SKR226 | NGB101304 | WBH 1304-7 | | | | 163 | 22.1 | 23.6 | |
| SKR227 | NGB101304 | WBH 1304-8 | | | | 154 | 21.4 | 20.9 | |
| SKR228 | NGB101304 | WBH 1304-9 | | | | | 14.5 | | |
| SKR229 | NGB101304 | WBH 1304-10 | | | | 102 | 21.3 | 26.5 | |
| SKR230 | NGB101836 | 1 | | | | 123 | 20.9 | 25.1 | |

Of the 227 accessions grown in the greenhouse, 199 were chosen for total nitrogen analyses based on previous information of high protein content. These were grown at the experimental farm of University of Copenhagen in Taastrup (55°40′ N; 12°18′ E), Denmark in the years 2017, 2018 and 2019. One individual per accession per year was analyzed. The seeds were hand-sown at approximately 1 seed per 10 cm in a two meter single row, and nets supported the growing plants. Both greenhouse plants and plants grown in the field were organized in the order of the Nordic Genbank number. Automatic irrigation was used when needed. At maturity, whole plants were harvested keeping accessions separate and then air-dried. Pods were collected from the dry plants and threshed by hand.

### 2.2. Microsatellites

Total genomic DNA was extracted from green leaf material of 194 pea accessions via the CTAB method [35]. The DNA was dissolved in 50 μL 1× TE buffer and the quality of the total genomic DNA was assessed via agarose gel electrophoresis (Electrophoresis Consort EV265, Cohasset, MA, USA) and spectrophotometry on an Uvidoc (Buch & Holm, Herlev, Denmark).

The microsatellite markers AA5, AC58 and A9 were chosen after initial screening of a larger set of microsatellite markers and based on previous success reported by Loridon et al. [28]. When tested AA5, AC58 and A9 produced consistently good quality data across the large number of accessions. All forward primers had an M13-tail (CACGACGTTG-TAAAACGAC) with a dye attached. The forward primer of AA5 had a FAM, A9 a NED and AC58 a VIC dye (Table 2).

**Table 2.** The three microsatellite markers AA5, A9 and AC58 [28], the forward and reverse primer sequences of the markers, and the color attached to the M13-tail of the forward primer. The sequence of the M13-tail was: CACGACGTTGTAAAACGAC.

| Marker | Forward Primer (5′—3′) | Reverse Primer (5′—3′) | Color |
|---|---|---|---|
| AA5 | tgccaatcctgaggtattaacacc + M13 | cattttgcagttgcaatttcgt | FAM, Blue |
| A9 | gtgcagaagcatttgttcagat + M13 | cccacatatatttggttggtca | NED, Yellow |
| AC58 | Tccgcaatttggtaacactg + M13 | cgtcaatttctttatgctgag | VIC, Green |

Each PCR sample consisted of 1 μL total genomic DNA, 7.25 μL double-distilled water, 1 μL 10× Dreamtaq buffer (20 mM Mg Cl$_2$), 0.4 μL dNTP (2.5 μM), 0.125 μL forward primer with M13-tail and color (10 μM), 0.125 μL reverse primer (10 μM), and 0.1 μL Dreamtaq Polymerase. The samples were mixed and run on a T100 Thermal Cycler (Bio-Rad Laboratories, Copenhagen, Denmark) using the following protocol: 94 °C for 4 min, 19 cycles of 94 °C for 1 min, 64 °C for 30 s (annealing), and 1 min of 72 °C. The annealing temperature was decreased half a degree for every cycle ending at 55 °C. Then, an additional 19 cycles were conducted with the same steps using a constant annealing temperature of 55 °C, followed by 10 min at 72 °C, and a final cooling to 4 °C.

The PCR products were diluted 1:5, and 3 µL of the diluted PCR product was mixed with 10 µL formamide and 0.5 µL GeneScan™ 120 LIZ™ (standard). The samples were denatured at 95 °C for 5 min and put on ice for 2 min. The samples were applied to an ABI 3130xl Genetic Analyzer (Hitachi, Tokyo, Japan) using the program Genetic Mapper Instrument protocol Fragment36_POP7_G5, and the resulting fragments were visualized on GeneScan® 3.7 Analysis Software (Applied Biosystems, Roskilde Denmark).

The resulting microsatellite band patterns were treated as binary characters where the presence of a given PCR fragment was marked as 1, absence as 0 and lacking results were marked as "?". All data from the three microsatellite markers were combined into one Microsoft Excel file and later converted to a text file (.txt) to generate the full matrix.

### 2.3. Thousand Seed Weight and Protein Content

Thousand seed weight (TSW) was determined by seed counting on a Contador Seed counter (Pfeuffer, Kitzingen, Germany) and then weighed.

Total N was determined via the Dumas method on a Vario Macro Cube (Elementar) [36] and crude protein content (%, $w/w$) was calculated using the conversion factor 5.44 [37,38].

### 2.4. Data Analysis

#### 2.4.1. Polymorphism Information Content

Polymorphism information content (*PIC*) was calculated for the microsatellite data. The values were calculated using the following equation, where $p_i$ is the frequence of the $i$ allele [27] (see also [28]):

$$PIC = 1 - \sum p_i^2$$

#### 2.4.2. Neighbor Joining Tree

A Neighbor Joining tree was constructed based on the binary matrix in PAUP 4.0a169 [39]. The parameters were set to BioNJ method with mean character difference as distance measure, and all characters were given equal weight. A Neighbor Joining bootstrap analysis was run with 2000 repetitions. Bootstrap values of 50% or above were manually added to the Neighbor Joining tree.

#### 2.4.3. Principal Coordinate Analysis of Microsatellite Data

A Principal Coordinate analysis (PCoA) was performed in GenAlEx [40,41] on the microsatellite data. Binary data were converted into columns of codominant genotypic microsatellite data with loci scored as fragment size in accordance with the GenAlEx-formats. A genetic distance tri-matrix was then made in GenAlEx, from which a PCoA was performed.

#### 2.4.4. ANOVA of Crude Protein across Years

A one-way ANOVA was performed in Microsoft Excel, comparing the difference between protein content across the three years. A Tukey–Kramer post hoc test was performed in Excel to compare the mean between each pairwise combination of years.

## 3. Results

Single seed descent was used to obtain a homogeneous seed stock of 227 genebank accessions for further field evaluations and for DNA extractions, molecular identification and phylogenetic analysis.

### 3.1. Plant Material

All seeds germinated and leaf material from all plants were collected for DNA extraction.

### 3.2. Microsatellites

DNAs were successfully extracted from all 194 samples selected for microsatellite analyses. PCRs from the marker AA5 resulted in 185 successful amplifications (95.4%), marker AC58 and A9 both resulted in 186 (95.8%) successful amplifications (Table 1). All together, a total of 25 accessions lacked results for one of the three primer pairs: SKR 8, 27, 42, 78, 97, 99, 100, 103, 106, 110, 117–119, 123, 125, 129, 131, 134, 135, 138, 182, 200, 202, 216 and 218 (Table 1).

The final matrix included 77 different PCR fragments from the three primer pairs. Marker AA5 provided 21 PCR fragments, marker AC58 27 PCR fragments, and marker A9 had 29 PCR fragments. The length of the PCR fragments varied from 110 to 407 base pairs (bp), where the smallest fragments were from AA5 and the largest from A9.

For the marker AA5, 5 of 21 PCR fragments were uninformative. The most common PCR fragment for AA5 was present in 64 samples and the length was 127 bp. For the marker AC58, 9 of 27 PCR fragments were uninformative. The most common in AC58 was present in 37 samples. For marker A9, 9 of 29 PCR fragments were uninformative. In A9, the most frequent PCR fragment was present in 26 accessions and had the length of 387 bp.

### 3.3. Thousand Seed Weight and Protein Content

Thousand seed weight (TSW) was determined for 102 field grown accessions in 2017 as a measure of seed size and ranged from 64 to 254.2 g (Table 1; Figure 1) with a mean of 152.05 g. The distribution (Figure 1) indicated that the majority of the accessions had a TSW between 120 and 180 g.

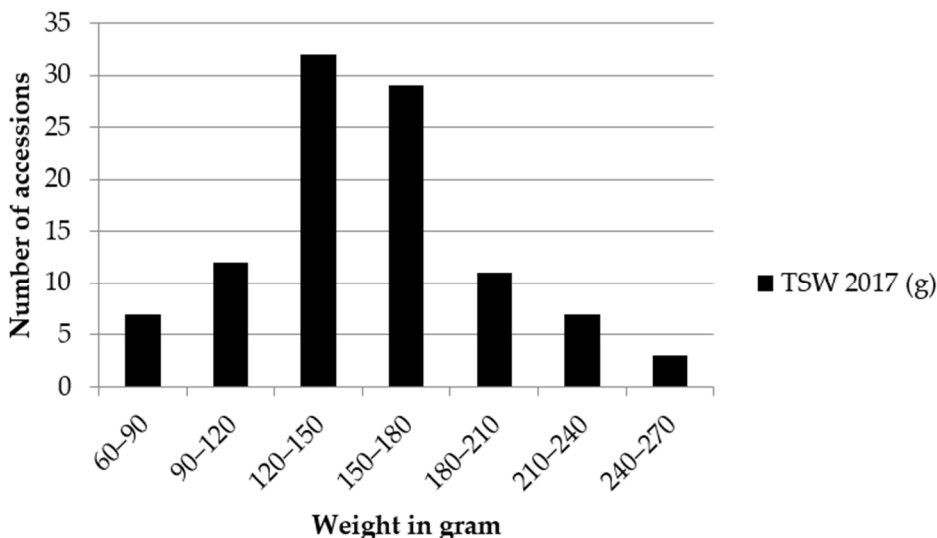

**Figure 1.** Thousand seed weight of pea accessions in grams (n = 102). The peas were grown with irrigation and standard agricultural management in 2017 and harvested at maturity.

The protein content was successfully determined for field grown material, with 199 accession in 2017, 95 accessions in 2018 and 30 accessions in 2019 (Table 1). The protein content varied from 9.3% (SKR 182, NGB102130) to 34.1% (SKR 64, NGB105862) for all years (Table 1) and most samples had a protein content between 16% and 28% protein content (Figure 2). All years showed a Gaussian distribution of protein content of the accessions. The average was 20.7% in 2017, 26.2% in 2018 and 22.7% in 2019, and 13 accessions had a protein content above 28% (five above 30%) in 2017, 19% (five above 30%) in 2018 and 0 in 2019. Of the 10 accessions with the highest protein content in 2017, only 1 was in the top ten in 2018, although the 95 pea accessions selected for 2018 generally had a higher protein content and were chosen among the 199 accessions with the highest protein content in 2017. One-way ANOVA revealed a highly significant difference in protein content across

the three years ($F_{2.21}$ = 66.05, $p$ = 9.44 × 10$^{-25}$). Tukey–Kramer post hoc test found that the protein content was significantly different among years.

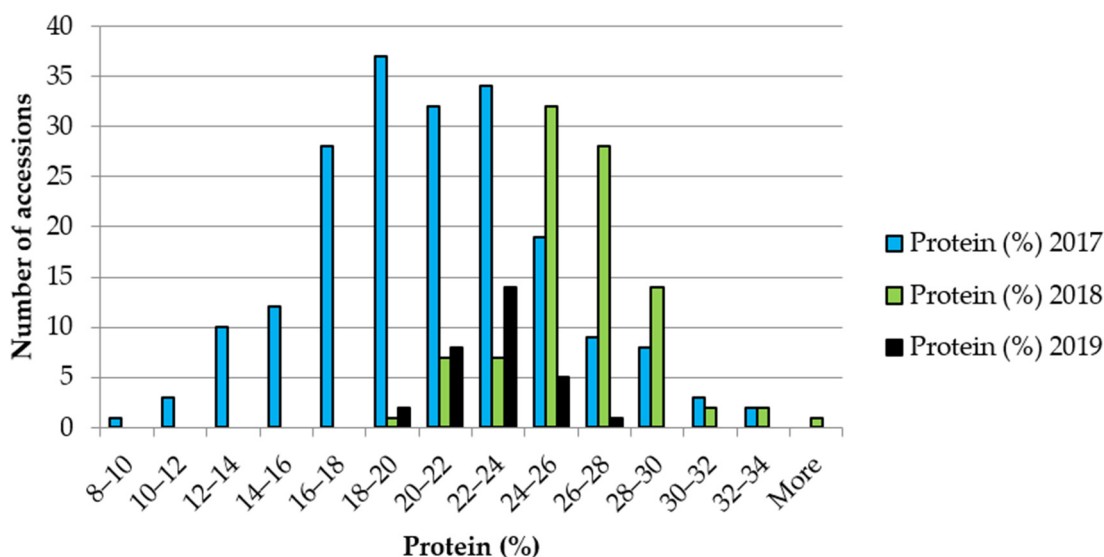

**Figure 2.** Protein content (%, $w/w$) of pea accessions. Blue represents accession were from 2017 (n = 199), Green from 2018 (n = 95) and Black 2019 (n = 30).

### 3.3.1. Polymorphism Information Content

The *PIC* values for marker AA5, AC58 and AA9 were 0.87, 0.90 and 0.91, respectively. The average *PIC* value was 0.89.

### 3.3.2. Neighbor Joining Tree

The Neighbor Joining analyses resulted in a tree topology with 107 groups of 194 accessions (Figure 3). The bootstrap analysis resulted in branch support for 11 groups ranging between 50% and 89%. Most groups did not have bootstrap support (Figure 3). We had cultivar names for 77 of the 194 accessions included in the NJ analysis. Fifty-three cultivars were represented by only one accession. Twelve accessions were from the cultivar *Pisum sativum* "chlorotica", four from "variomaculata", two from "compactum", two from "costata", two from "New Wales", and two from "Lilaska" (Figure 3; Table 1). The "chlorotica" cultivars were resolved in several different groups on the NJ tree, but most of them clustered around the same few branches (Figure 3). The other cultivars resolved far apart on the NJ tree.

### 3.3.3. Principal Coordinates of Microsatellite Data

The first two principal components of the PCoA accounted for 16.62% of the variance between the accessions of the three microsatellites combined (Figure 4). The pattern of variation mirrored the placement of samples in the NJ tree.

### 3.3.4. Nitrogen Content and Thousand Seed Weight in Relation to Genetic Diversity

The results from the total N analyses showed no apparent relationship between the protein content and the genetic diversity among the pea accessions. The accessions with high protein content did not cluster either in the NJ tree or in the PCoA.

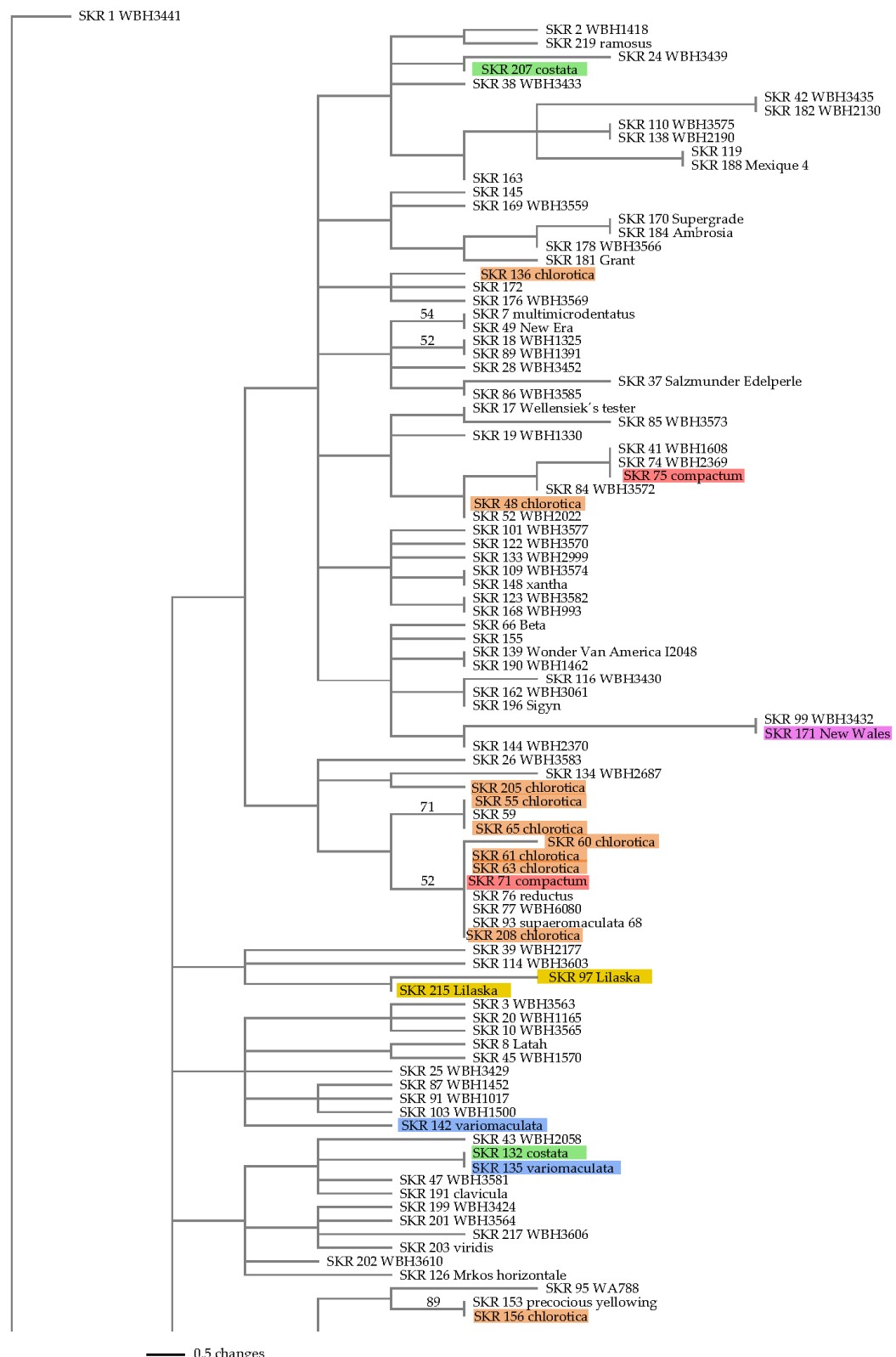

**Figure 3.** *Cont.*

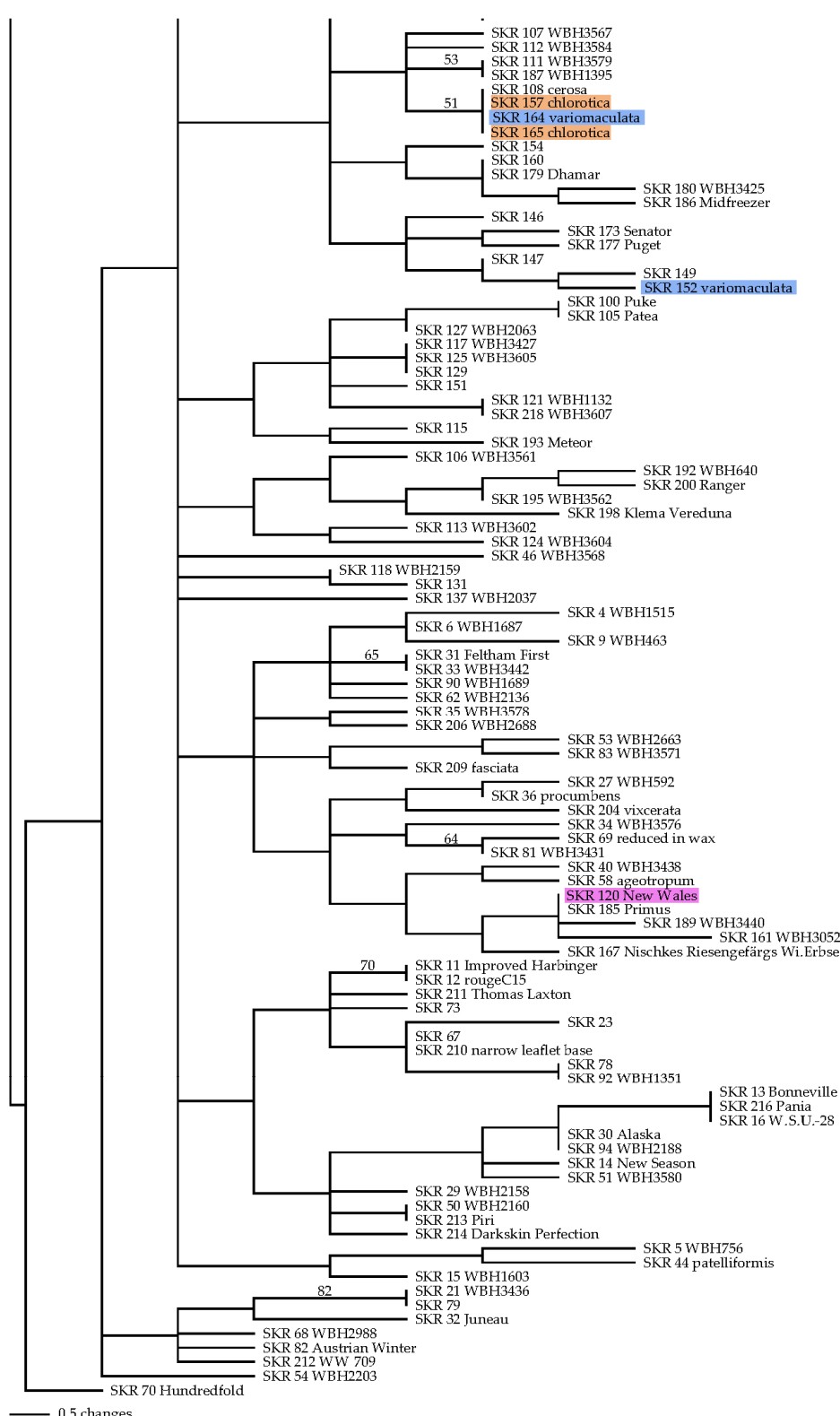

**Figure 3.** Neighbor Joining tree of 194 pea accessions with bootstrap values on supported branches. Highlighting indicates cultivars represented by more than one accession. Green: "costata"; orange: "chlorotica"; red: "compactum"; purple: "New Wales"; yellow: "Lilaska"; blue: "variomaculata".

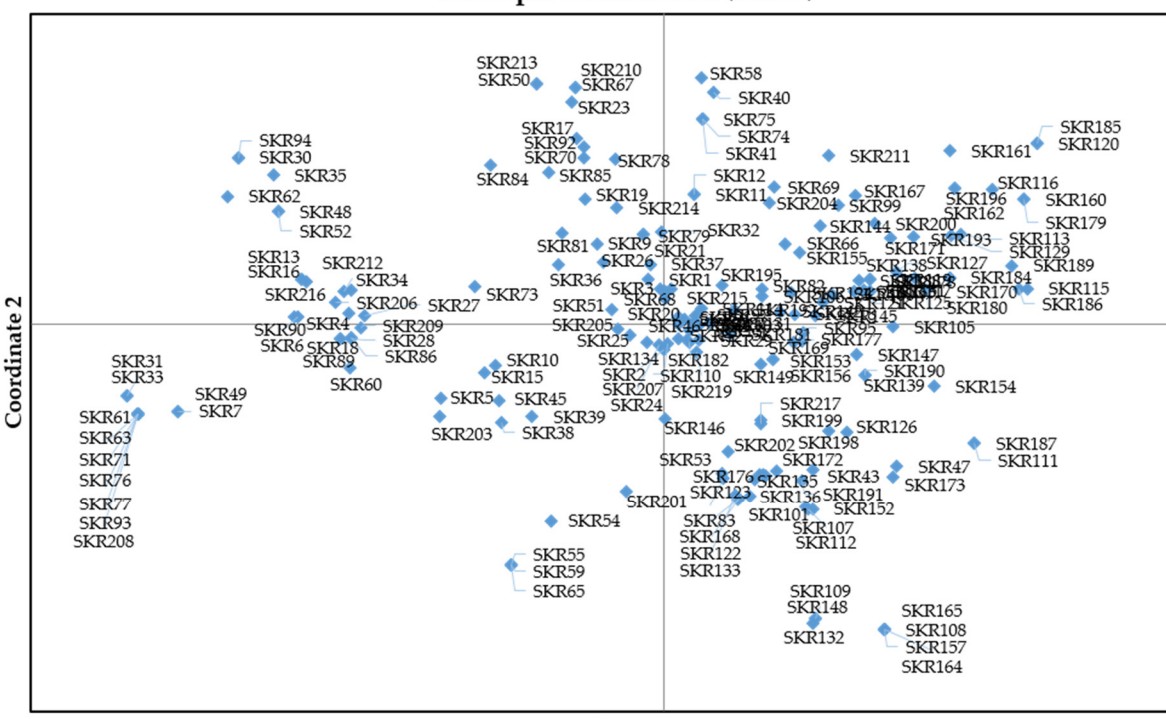

**Figure 4.** Principal coordinates analysis of the microsatellite results. X- and y-axis represent PC1 and PC2, respectively.

## 4. Discussion

The primary goal of this study was to examine the genetic diversity in pea accessions, which have primarily been grown and collected in Scandinavia. The genetic variation was investigated using three microsatellite markers, which all functioned well and resulted in high genetic variation. Furthermore, the protein content was determined for different subsets of the accessions over three years and thousand seed weight was measured for one year.

### 4.1. Microsatellites

The microsatellite AC58 produced 186 PCR fragments that varied between 210 and 263 bp. The number of fragments were higher than other studies where 1–11 fragments are reported, but the lengths were within what other studies have found: 200–263 bp [15,19,28,31,34]. The microsatellite AA5 produced 185 different PCR fragments of lengths from 110 to 264 bp. The number of fragments were higher than other studies where 1–3 fragments are reported, and the lengths spanned a wider range than the 225 to 250 bp reported in other studies find [15,28,34]. The microsatellite A9 produced 186 different PCR fragments with lengths that varied between 325 and 407 bp. The number of fragments were higher than in other studies, where one to six fragments were reported, and the lengths spanned a somewhat wider range than the 330–390 bp observed in previous studies [14,17,18,21,28,31].

To sum up, most previous studies produced 2–4 DNA fragments for each microsatellite, except for the study of Hagenblad et al. [19], where 14–33 DNA fragments were found. Hagenblad et al. [19], like in our study, used Scandinavian pea accessions from Nord-Gen. This indicates that the collection of Scandinavian pea accessions at NordGen are genetically diverse. The results could have been even more discriminating if more than three microsatellites had been analyzed.

### 4.2. Polymorphism Information Content

The *PIC* values in this study (0.87–0.91) were high when compared to other studies. The reported *PIC* values for pea accessions fall within a wide range from 0.3 to 1.0 for individual microsatellite markers. The variation in *PIC* values is higher in studies developing microsatellites, e.g., *PIC*s varied between 0.04 and 1.0 with an average of 0.63 in Loridon et al. [28]. Many studies focus on the genetic diversity within a specific group of accessions and use already tested microsatellites. The *PIC* values of these studies do not span the same variation as Loridon et al. [28] did. A few studies did find *PIC* values as high as in this study. Burstin et al. [13] use a large number of accessions and many microsatellites and obtained *PIC* values between 0.46 and 0.97 with a mean of 0.8 for 372 wild and cultivated pea accessions using 29 microsatellites. Nasiri et al. [33] found an average *PIC* value of 0.72 for 77 pea accessions from 17 different countries with 10 microsatellites. However, most studies found lower *PIC* values than our study, between 0.3 and 0.65, most using a larger number of microsatellites [14,16–18,20–24,28,30–32]. Singh et al. [20] use 20 microsatellites to assess the genetic diversity of 47 pea accessions from India and obtained an average *PIC* value of 0.55, ranging from 0.04 to 0.85. Another study [22] assessed 40 pea accessions with five microsatellites and obtained *PIC* values between 0.14 to 0.82. Of all studies assessing genetic variation in peas [13–26,29–34], only three studies [13,15,33] reported results where all the microsatellites had a *PIC* value of 0.46 or higher.

The *PIC* values are mainly affected by the variation among the included individuals; however, the number of microsatellites may affect the outcome of the average *PIC* value because the number affects the statistical strength by potentially adding more variation [27,31]. The genome of *Pisum sativum* is large, approximately 4.45 giga base pairs [3], and therefore the results of this study could be affected by the low number of microsatellites included three.

The *PIC* values in this project varied between 0.87 and 0.91 with an average of 0.89. For the same specific microsatellites, Loridon et al. [28] finds *PIC* values of 0.69 (A9), 0.84 (AC58) and 0.78 (AA5). Other studies using the A9, AC58 and AA5 microsatellites find *PIC* values between 0.61 and 0.78 [14,15,17–19,31,34] except for the A9 microsatellite of Haliloglu et al. [17], which has a low *PIC* value of 0.03.

The *PIC* values of this study were therefore among the highest reported for genetic variation among pea accessions [13–26,28–34]. The average *PIC* value is not only affected by the variation among individuals for each microsatellite included but also by the number of accessions, because the chance of identifying different alleles rises with the number of accessions included [27,31]. Thus, the higher *PIC* values in the present study could partially be explained by the larger number of accessions (194) compared to most studies [16–23,26,28,29,31–34]. Some studies included more than 100 accessions [13–15], however, and few more than 500 [24,25,30].

This study, along with most of the studies on genetic diversity of peas, placed the microsatellites used in the category, "highly informative", because the *PIC* values are above 0.5 [27]. The high *PIC* values of this and other studies could be explained by the selection of the most polymorphic markers from the study of Loridon et al. [28] or similar studies where a large number of microsatellites were included. The high average *PIC* value of this study could be interpreted as a reflection of a high genetic variation in general among the 194 accessions included. This high genetic variation indicates that the accessions had not undergone a targeted breeding for specific agronomical traits and homozygosity [32]. The high *PIC* values might thus be explained by the inclusion of a large number of pea accessions with different breeding histories.

### 4.3. Neighbor Joining Tree

The Neighbor Joining tree uniquely identified 153 of the 194 accessions (78%) and 108 groups were resolved as monophyletic using three microsatellites (Figure 3). Jain et al. [32] uniquely identify all 96 individuals included in the study in the NJ topology, but they included 42 polymorphic markers, distributed throughout the genome. Many

studies using more microsatellites have produced a full resolution of the NJ tree [26,31,32]. One study using three microsatellites obtained 95% resolution of the topology, but this study was partly at the species level [29], whereas the present study was below the species level. In the study by Bouhadida et al. [30], 18 of 19 accessions (94.7%) are uniquely identified on an unweighted pair group method (UPGMA) topology through the use of five microsatellite markers. The UPGMA and NJ methods use the same underlying model, but UPGMA assumes a similar rate of evolution rate along the branches of the tree (molecular clock) [42]. The bootstrap support values of our study were low (Figure 3) [43]. The low bootstrap support and the lack of resolution in the NJ topology is mainly explained by a lack of variation due to insufficient DNA fragments of each microsatellite. A total of 77 variable DNA fragments is not enough to fully resolve 194 accessions [43]. Another possible explanation for the low bootstrap support is the taxonomic level. For studies below the species level, bootstrap values cannot always be expected to be high [43]. A third factor affecting the outcome of the NJ topology in this study is missing results for some individuals. Twenty-five accessions lacked data for one of the three microsatellites, which contributed to greater uncertainty in the branching pattern and collapse of branches [44].

The genetic diversity of named cultivars was expected to be structured such that accessions with the same cultivar name would be included in the same monophyletic group. However, such a structure did not appear. The 12 cultivars for which we had more than one accession were resolved in different positions on the NJ tree (Figure 3, colored cultivars). One group included accessions from "chlorotica", "reductus", "superaeromaculata" and "compactum" (Figure 3, colored cultivars). This lack of structure could partly be explained by the lack of sufficient data [3,24,34]. However, the resolution of individuals from the same cultivar at different places in the NJ tree may also be the result of genetic variation among individuals. This variation may be due to separation in time creating genetic diversity because of crosspollination resulting in polymorphism [29]. Peas from Ethiopia are genetically similar and resolve as a monophyletic group, whereas accessions from other countries, e.g., USA and Norway, do not show such similarity and do not resolve as monophyletic [33]. This lack of similarity can be explained by a gene flow between countries, e.g., use of genetic material from more than one country. The lack of a clear pattern among cultivars included in this study might be due to a high exchange of genetic material among breeding companies in Scandinavia, because the accessions of the same cultivar were not necessarily from the same source.

*4.4. Thousand Seed Weight and Protein Content*

Seed size is an important agronomic trait for yield and productivity that is inherited and characteristic of specific cultivars. Seed size is a quantitative trait controlled by many genes. Thousand seed weight is a simple method for measuring seed size. We found large variation from 64.4 g (SKR 46, NGB103568) to 251.2 g (SKR 218, NGB103607).

Protein content varied from 9.3% (SKR 182, NGB102130) to 34.1% (SKR 64 NGB105862) over all years and accessions. One goal of this study was to identify high-protein pea accessions, and 13 were found to have a protein content above 28% in 2017 and 19 in 2018, but none in 2019. This emphasizes the variation in protein content across the years, as the accessions with determined protein content for the years following 2017 was selected for high protein content. The selection of peas with a high protein in 2017 for analyses in 2018 moved the distribution to the right (Figure 2). We expected the protein content of the 2019 samples to show the same pattern and be placed even further to the right, but this was not the case. Year to year variation in protein content is well documented and is assumed to be due to differences in weather [45]. In comparison to other studies, the protein content was 23.9 ± 2.5% and TSW 195.9 ± 16.9 g for a collection of 1222 accession of cultivars and landraces cultivated during 1979–1982 with irrigation and standard management in Sweden [46]. A recent study of 50 accessions assessed for their potential in the Arctic region showed an average protein content of 26% in Denmark, 24% in Sweden and 20% in Finland [47].

An analysis of 198 peas (*Pisum sativum* L. cv. Trapper) from Saskatchewan farmers showed a protein content of 14.5% to 28.5% with a Gaussian distribution [11]. The protein content was determined from the Kjel-Foss method with a conversion factor of 6.25. Wang and Daun [10] determined the protein content via the Dumas method (N × 6.25) for six samples of four different pea varieties with a crude protein content ranging from 20.2% to 26.7%. Thus, the studies show that there is natural variation among peas. The variation in our study was more wide-ranging, and while some accessions had a lower protein content than that reported in previous studies, there were accessions that were higher than in most other reports. Only Slinkard [48] reported a protein content above 30%.

### 4.5. Future Perspectives

The next step could be to investigate the history of and characterize the cultivars and accessions included in this study. These accessions were primarily from Scandinavia and all from Nordic Gene bank and they were chosen primarily because there were some indications but not necessarily confirmations of high protein content.

A high genetic diversity among pea accessions from a gene bank is valuable because researchers want to preserve as much diversity as possible for future breeding. By targeted crossing of genetically diverse individuals, a selection of plants with desirable traits such as high protein, high resistance or tolerance to drought, followed by breeding towards homozygosity, it is possible to actively use the preserved genetic diversity to develop pea cultivars for the future [3,12,32].

Microsatellites are only one of several methods to study genetic diversity. Microsatellites are popular due to their co-dominance and high polymorphism. Methods in which differences across the genome are targeted give more data and are more likely to resolve genetic diversity among accessions. An example of such a method is Diversity Array Technology (DArT), [49].

Our results increase our understanding of genetic diversity across pea varieties, and could contribute to the sustainable cultivation of peas and adaptation to climate change.

## 5. Conclusions

Homogeneous lines of 227 pea accessions from seed stocks at NordGen were produced via single seed descent. The peas were grown in a field for 2 years and the seed protein content showed a variation between 9.3% and 34.1% over years and accessions and thousand seed weight were on average 152.05 g. More than 10 accessions had a protein content above 28%, showing that the collection has potential as breeding nursery for high-protein pea cultivars. Three microsatellites were highly informative. The microsatellites separated 153 of 194 accessions and had *PIC* values between 0.87 and 0.91, indicating that the gene bank material contains a large number of pea accessions with different breeding.

**Author Contributions:** Conceptualization, S.K.R., G.P. and C.B.A.L.; methodology, S.K.R. and C.B.A.L.; software, L.W. and C.B.A.L.; validation, C.B.A.L. and L.W.; formal analysis, C.B.A.L., L.W. and S.K.R.; investigation, C.B.A.L., S.K.R. and L.W.; data curation, L.W. and C.B.A.L.; writing—original draft preparation, L.W.; writing—review and editing, L.W., S.K.R. and C.B.A.L.; visualization, L.W. and C.B.A.L.; supervision, C.B.A.L. and S.K.R.; project administration, S.K.R.; funding acquisition, S.K.R. All authors have read and agreed to the published version of the manuscript.

**Funding:** This research was funded by The Danish Agricultural Agency, Genetic Resources, grant numbers 16-3262-000061 and 18-26051-000011; and organic RDD Peas & Love project 34009-21-1895.

**Data Availability Statement:** Data are available from the Dryad Digital Repository: at https://doi.org/10.5061/dryad.d7wm37q6b, accessed on 29 August 2023.

**Acknowledgments:** We thank NordGen for providing seeds of pea accessions, Didde Hedegaard Sørensen and Vinnie Deichmann for support in the lab, Weiyao Fan for help with the field trails, the anonymous reviewers for helpful comments on the manuscript, and Jacob Weiner for editorial support.

**Conflicts of Interest:** The authors declare no conflict of interest. The funders had no role in the design of the study; in the collection, analyses, or interpretation of data; in the writing of the manuscript, or in the decision to publish the results.

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
