# Peer review of "Assessment of Genetic Diversity and Protein Content of Scandinavian Peas (Pisum sativum)"

_agronomy, doi:10.3390/agronomy13092307_

Round 1
Reviewer 1 Report (Previous Reviewer 4)
This is an excellent paper. The added materials and changes show the originality and quality of this research.
I see no major problems with the English.
Author Response
Please see actions in the attached file.

Reviewer 2 Report (New Reviewer)
The manuscript, “Assessment of Genetic diversity and protein content of Scandinavian peas (Pisum sativum)“, describes the genetic diversity of a collection of 227 Scandinavian pea accessions from the Nordic Genebank using three SSR markers. Field trials were conducted to access thousand seed weight and protein content of the collection. While theses are information that are generally of interest for breeders who intend to utilize genetic resources from genebanks, this study cannot quite deliver what it promises due to the low number of markers. This leads to a poor resolution of the genetic relationship, even when the presumed close relationship of the accessions is taken into account. In order to access the genetic diversity in this collection properly, more SSR markers are required. Alternatively, different markers may be appropriate; e.g. SNPs created by some sort of reduced representation NGS. Another issue of this manuscript is the sometimes wrong or imprecise statements in the discussion.
For some more detailed notes, see below:
L30
“very large genome” is a relative statement. Since the exact size is known, use it here.
L40-54
One of the questions that arise when reading this manuscript is, why microsatellites and not more modern marker types were used. This would be a good place to add some information on the advantages of microsatellites compared to other marker types; e.g. cost efficiency
L102-109
What was the experimental design of the field trials?
L266
What variation and what is “high”? High relative to what?
L272
It would give that paragraph more substance, if the relationship of genetic and phenotypic variation had been tested statistically.
L280
What does “good results” mean here?
L297
How does Hagenblad et al. compare to this study?
L331-332
“The PIC value is not only affected by the number of microsatellites included but also by the nuber of accessions…”
This is not quite correct. The PIC is not affected by the number of microsatellites; also see L 321.
The number of individuals is also not the main driver of the PIC, but rather the variation among the included individuals.
Both is apparent from the formula of the PIC.
L342-343
This is an overstatement. While it is true that the PIC rises with increased allelic diversity, it is not clear which and how many accessions harbor this diversity. Moreover, three microsatellites are not sufficient to make a statement on the diversity of 194 accessions.
L365-367
That is not true. There are numerous studies of domesticated species, including domesticated pea, with high bootstrap values. The reason for the poor support and resolution of the NJ tree, as indicated by the large number of polytomes and the low bt node support, is the low number of markers.
Note also some inconsistencies of the citing style throughout the manuscript, e.g. L63
Author Response
Please see actions taken in the attached file. Many thanks for the thorough review.

Reviewer 3 Report (New Reviewer)
In this study, the authors investigated the genetic diversity of pea accessions which primarily been grown and collected in Scandinavia using three SSR markers and found the average PIC value was 0.89. The authors provided field experiment data for seed protein content and thousand seed weight. Interestingly, several accessions had protein content above 28%, showing that the collection has potential for high-protein pea breeding. I suggest acceptance of the manuscript after the minor reversion.
Minor concerns:
1. Why did the authors use only three markers including A9, AC58 and AA5 for microsatellite analysis?
2. The authors should list the accession information inculding13 accessions had seed protein content above 28% (five above 30%) in 2017, 19% (five above 30%) in 2018 in a new table.
Author Response
Please see the actions taken in the attached file.

Round 2
Reviewer 2 Report (New Reviewer)
(line numbers correspond to the original manuscript and the authors' response file)
L102-109
Block design, completely randomized or anyhing else, how many repetitions, what was the experimental unit (e.g. the two meter rows, individuals), ...?
L331-332
Once again, the number of markers does not affect the PIC. The PIC only depends on the allele frequencies and, hence, also on the number of alleles, but not the markers. How could it be affected by the number of markers? Each marker has it's own PIC.
The average PIC is affected by the number of marker, but that is trivial.
Author Response
The reviewer has two comments1) L102-109. Block design, completely randomized or anyhing else, how many repetitions, what was the experimental unit (e.g. the two meter rows, individuals), ...?
Responce: The lines 102-109 does not line up with the comment. The reviewer must mean line 105 - xxx, otherwise it makes no sense. Lines copied in below and highlighted in yellow in the text.
Of the 227 accessions grown in the greenhouse, 199 were chosen for total nitrogen analyses based on previous information of high protein content. These were grown at the experimental farm of University of Copenhagen in Taastrup (55°40′ N; 12°18′ E), Denmark in years 2017, 2018 and 2019. One individual per accession per year was analyzed. The seeds were hand sown approximately 1 seed per 10 cm in a two meter single row, and nets supported the growing plants. Both greenhouse plants and plants grown in the field were orgainzed in the order of the Nordic Genbank number. Automatic irrigation was used when needed. At maturity, whole plants were harvested keeping accessions separate and then air dried. Pods were collected from the dry plants and threshed by hand.
We have added more description according to the reviewer. See track changes in the manuscript. However, other reviewers did not agree with this reviewer.
L331-332
Once again, the number of markers does not affect the PIC. The PIC only depends on the allele frequencies and, hence, also on the number of alleles, but not the markers. How could it be affected by the number of markers? Each marker has it's own PIC.
The average PIC is affected by the number of marker, but that is trivial.
Responce:
We are in complete agreement with the reviewer and had corrected the text accordingly. We now have added the word "average" to the the text to be super extra clear.
This manuscript is a resubmission of an earlier submission. The following is a list of the peer review reports and author responses from that submission.
Round 1
Reviewer 1 Report
Winther et al presents a preliminary study on exploration of genetic diversity in an important set of pre-breeding accessions in Nordic Pea collection. The manuscript is well-written and results are appropriately presented except some common English language mistakes.
It would be nice to see more detailed information about the accessions OR then cite a proper database (if exists) to make readers and probable future studies to access this material based on some laid criteria (such as location / pedigree etc).
A PCA analysis would be appreciated by readers to show if some of the accessions cluster together.
NJ Tree diagram is low quality, please provide a better quality figure in the revised version.
Please supplement PCR pictures
Reviewer 2 Report
The manuscript is clear and the results well presented. I have no major remarks to make. The results could have been even more discriminating by analysing more than 3 SSRs.
L26: put Latin names in italics
L29: add a space between "(4).Pea".
L53: delete a "."
L64: add a space between "(29).In"
Suggestions for discussion:
1. figure 2 suggests a "year" effect in the protein content. You could discuss this aspect in the result and discussion section.
2. L262: "4.2 Thousand seed weight...". I would put this paragraph after the paragraphs on genetic analysis. 4.1. Microsat; 4.2. PIC; 4.3.NJT; 4.4 thousand seed...; 4.5. future perspectives.
Reviewer 3 Report
The manuscript entitled “Assessment of Genetic diversity and protein content of Scandi- 2
navian peas (Pisum sativum) ” describes the genetic diversity of 227 pea accessions from Nordic Gene
bank using three microsatellites markers, A9, AC58 and AA5. The high PIC values between 0.87 and 0.91 indicated that the gene bank material contains large number of pea accessions with different breeding history.
The authors also grew the peas in the field for two years and showed variation in the seed protein content between 9.3% and 34.1% over years and accessions.
Although the authors only had 10 accessions with protein content above 28% the collection has potential as breeding nursery for high-protein pea.
Therefore, I recommend this manuscript for publication in ‘Agronomy’.
Reviewer 4 Report
This paper appears to be prepatory work towards a protein breeding program in peas. It does not present novel insights or utilize high quality methods.